# 3D Printing in Development of Nanomedicines

**DOI:** 10.3390/nano11020420

**Published:** 2021-02-07

**Authors:** Keerti Jain, Rahul Shukla, Awesh Yadav, Rewati Raman Ujjwal, Swaran Jeet Singh Flora

**Affiliations:** 1Department of Pharmaceutics, National Institute of Pharmaceutical Education and Research (NIPER)—Raebareli, Lucknow 226002, India; keertijain.02@niperraebareli.edu.in (K.J.); rahul.shukla@niperraebareli.edu.in (R.S.); awesh.yadav@niperraebareli.edu.in (A.Y.); 2Department of Pharmacology and Toxicology, National Institute of Pharmaceutical Education and Research (NIPER)—Raebareli, Lucknow 226002, India; ujjwalr.raman@gmail.com

**Keywords:** nanomedicine, 3D printing, nanomaterials, implants, tissue engineering

## Abstract

Three-dimensional (3D) printing is gaining numerous advances in manufacturing approaches both at macro- and nanoscales. Three-dimensional printing is being explored for various biomedical applications and fabrication of nanomedicines using additive manufacturing techniques, and shows promising potential in fulfilling the need for patient-centric personalized treatment. Initial reports attributed this to availability of novel natural biomaterials and precisely engineered polymeric materials, which could be fabricated into exclusive 3D printed nanomaterials for various biomedical applications as nanomedicines. Nanomedicine is defined as the application of nanotechnology in designing nanomaterials for different medicinal applications, including diagnosis, treatment, monitoring, prevention, and control of diseases. Nanomedicine is also showing great impact in the design and development of precision medicine. In contrast to the “one-size-fits-all” criterion of the conventional medicine system, personalized or precision medicines consider the differences in various traits, including pharmacokinetics and genetics of different patients, which have shown improved results over conventional treatment. In the last few years, much literature has been published on the application of 3D printing for the fabrication of nanomedicine. This article deals with progress made in the development and design of tailor-made nanomedicine using 3D printing technology.

## 1. Introduction

Three-dimensional (3D) printing innovation has developed much over the last 20 years. It has helped in providing suitable information for the manufacturing process of various industries including pharmaceuticals, medicine, ecological monitoring, aviation, and automobiles, and as well as in research [1]. Charles Hull developed the 3D printing technology in 1986, as the process known as stereolithography (SLA) [2]. The layer-upon-layer manufacturing technology is the basis for 3D printing technology, which is based on development of 3D structures designed with the support of computer-aided design (CAD) drawing. Three-dimensional printing methods are very creative and ascended as an adaptable technology stage. Three-dimensional printing unties new probabilities and ideas for industrial manufacturing to improve efficiency and cost efficacy. Various materials such as thermoplastics polymers, graphene-based materials, and metal are the components used in 3D printing applications [3]. The progress made in the field of 3D printing technology have enabled researchers to explore different possibilities in the medical field, including fast growing areas, for example, drug delivery systems, tissue design, tissue and organ models, prosthetics and replica manufacture, inserts, and more. Three-dimensional printing is fast growing and an expanding technology within biomedical and pharmaceutical markets [4]. The different approaches and techniques used in 3D printing have been reviewed comprehensively by us and other scientists, previously, which we have not discussed in this article [5,6,7].

Three-dimensional printing is being explored to manufacture oral dosage forms of different geometry and size. The starting material for the 3D screen printing process was developed for controlled release of the model drug paracetamol (acetaminophen). The model screen print unit was used to manufacture different tablets in a single production process. The tablets were manufactured in three different sizes and five designed geometries, including disk, doughnut, oval, grid, and cuboid. The study of size and mass of the individual tablets showed high uniformity inside the different groups of tablets. Evaluation of their physical parameters of formulation, such as breaking force, friability, yield value, was found to be superior in comparison to conventional tablets. The study of drug release tests performed in artificial gastric media showed that paracetamol release depends on surface-to-area volume ratio. Studies of 3D printing enable potential manufacture of complex oral dosage forms for mass production with more reproducibility [8]. Thermoresponsive polymer poly(*N*-isopropylacrylamide) (PNIPAAm)-co-acrylic acid hydrogel was documented and effectively used for different kinds of 3D printing, including 3D single nozzle expulsion printing, 3D coaxial printing, and 3D hybrid printing. Three-dimensional printing with hybrid bioink using three different types of skin-related cells showed relatively high cell viability regardless of cell seeding density, printed position, or cultivation time. The bioengineered skin made from multilayered human umbilical vein endothelial cells (HUVECs), 3T3-J2 and HaCaT cells, showed external surface keratinization of the epidermis layer, and sprouting and splitting of the subcutaneous endothelial cells after in vitro air–liquid-interface induction. It is believed that the bioengineered skin-grafts might be used as potential implants for wound healing [9,10]. Three-dimensionalprinting-based design of perfusable vascular constructs using crosslinkedbioink is shown diagrammatically in Figure 1.

## 2. Nanomedicine and 3D Printing

Nanomedicine is the application of nanotechnology in various field of medicine such as site-specific drug delivery, theranostic tools, and in drug repurposing. Nanomedicine is a relatively recent field, extensively being explored for research and innovative ideas by scientists of different disciplines. Nanomaterials are also being used in combination as a hybrid to explore new possibilities such as dual targeting, stimuli responsive release, and theranostic applications. [11,12]. Nanomedicines are a promising tool for various theranostic application in infectious diseases, noncommunicable fatal diseases such as brain tumors, multidrug resistance (MDR) tumors, infectious diseases, metastatic tumors and relapsed tumors that are highly challengeable for clinical studies, crossing of therapeutic agents to brain (BBB), dodging the MDR pathways, inhibiting migratory tumor cells, immunostimulation, antiangiogenic activity, and removal of the cancer stem cells. Various nanotechnology, including nanoparticles (lipid nanoparticles, polymer nanoparticles, metallic nanoparticles), dendrimers, nanodiamonds, nanocrystals, quantum dots, carbon nanotubes, nanogels, nanoemulsions, etc., due to their extra small size with confined size distribution, are being explored for designing various nanomedicines. These nanomedicines are also being fabricated using 3D printing technology as precision medicines to satisfy individual specific needs by using apparent surface functional design along with bioconjugation and extreme biocompatibility [13,14,15,16,17,18].

Three-dimensional printing provides the ease in preparation and tailoring of micro- and nanoparticles conjugated with functional antibodies in a continuous mixing process with desired features [19]. Pal and coworkers fabricated a bilayer composite for tissue regeneration application using a 3D printed layer and nanofibrous layer. These composites are composed of a dual-pore system including a 3D printed matrix and nanofibrous layer with a pore diameter of ~200 and ~20.59 µm, respectively. The structural and compositional features conferred the composites with moderate hydrophobicity, providinga contact angle for the nanofibrous layer and 3D printed layer of 64.4° and 92.2°, respectively. Additionally, prepared composites showed a tensile strength of 6.12 ± 1.26 MPaand high water uptake capacity of ~95%. These parameters depicted the migration, adhesion of fibroblast cell, and observation of growth on either surface of the scaffolds [20].

Ceylan and coworkers designed a 3D printed enzymatically biodegradable microswimmer, based on hydrogel, for theranostic application. This magnetically powered microswimmer was fabricated using gelatin methacryloyl and biofunctionalizedsuperparamagnetic iron oxide nanoparticles (SPIONs). The scientists also tested this system in ErbB2 overexpressing SKBR3 breast cancer cells, where magnetic contrast agent and SPIONs, tagged with anti-ErbB2, were released rapidly due to enzymatic biodegradation of the microswimmer [21] (Figure 2). Similarly, multiphoton lithography-based 3D printing methodology was employed to fabricate carrier-free 3D shape-designed antigen nanoparticles (ANP). These 3D shape-designed ANPs were of distinct aspect ratios and could elicit shape-dependent immune responses [22] (Figure 3). Three-dimensional printing can be explored convincingly in the design of safe and effective precision medicines and nanomedicines, which are complex and difficult to be formulated with conventional top-down and bottom-up approaches.

## 3. Implication of 3D Printed Nanomedicines in Designing of Precision Medicines

Precision medicine is a practice for the treatment and prevention of disease and it studies the effect of genetics, environment, and life style on the requirement of treatment protocol for a particular patient. A precision medicine approach would mainly focus on each patient’s one-by-one characteristics for both analysis and treatment [23,24]. In January 2015, President of United States of America in one of his fine addresses gave emphasis and coined the term precision medicine, and correlated it with blood transfusion for making tailored interventions. Precision medicine is an advanced process to deal with tailored care and provides the accurate treatment at the right time to an individual [25]. Precision medicine is an innovative way to customize care that is focused on an individual’s characteristics, and to deliver to an individual correct treatment with correct timing [26].

In bone tissue engineering, a scaffold having biocompatibility and acceptable physiochemical properties is manufactured for improving bone properties with surrounding tissues. Three-dimensional printing innovation is being explored significantly to create advancement in the orthopedics field by providing precise scaffold as per the requirement of individual patients for bone tissue engineering. It increases bone development for cell culture to optimize osteogenesis and that is why bone tissue engineering is presently becoming feasible in today’s clinical field [27].

Viruses are threat to society from the past, with examples of Nipah virus, Ebola virus, hantavirus, and severe acute respiratory syndrome coronavirus 2 (SARS-COV-2). Recently, the COVID-19 pandemic caused by coronavirus SARS-COV-2 resulted in shortage of much personal protective gear. Hereby, 3D printing opened the horizons to make and develop or create necessary new equipment. Three-dimensionalprinting in the medical profession has emerged as powerful tool for combating coronavirus in surgical as well as critical areas with personal protective gear that fall into short supply during the COVID-19 pandemic crisis. One of best options for manufacturing a ventilation system, together with its connectors and sampling, surgical head and light accessories, face shields, mask clips, and swabs, was provided by 3D printing technology when worldwide supplies were disrupted [28,29,30]. Three-dimensionalprinting (Table 1) played a vital role in the medical field during the spread of COVID-19. The crisis arising from COVID-19 resulted in limited supply and facilitation of innovative ways to find tentative solutions to cater the emergency and unmet demand of patient care. Nonetheless, there still remains the requirement for introducing a reliable and safe means of 3D printing in medical facilities.

## 4. Three-Dimensional Printing in Development of Nanomedicines

The 3D printing system was developed by the Massachusetts Institute of Technology, Cambridge, MA 02139, United States in 1992. This new and prototype technology was used to design the constructs with the help of CAD. A terminal computer also plays an important role in manipulating the constructed models. A print head is used to spray the binding material. The spraying of material is governed by the X–Y orientation of the print head. A simple and rapid layer by layer process is adopted. The prototype is created on the powder and spread on the powder bed. The Z-vertical movement of the powder bed is controlled by a piston rod. The powder bed could be allowed to lower to achieve a specific thickness, and the process for printing is repeated several times to construct the proper design model. After the construction, the powder is detached and the design remains [35,36,37]. Different applications and approaches of 3D printing technologies are presented graphically in Figure 4A,B. In the following subsections, we discuss the various nanomedicines and precision medicines developed by using 3D printing. Basically, many nanomedicines are currently under development to be used as precision medicine, and 3D printing is also being investigated for the design of these nanomedicines.

### 4.1. Controlled Release System

A controlled-release drug delivery system allows the measured release of the active pharmaceutical ingredients over an extended time. A controlled drug release system is designed in a way that it ensures safety, enhanced effectiveness of the drug, and better patient compliance. In brief, a controlled drug release system governs the level of drug in the plasma with less repeated dosing. Various drug delivery systems, such asa controlled and targeted delivery system, fast disintegrating delivery systems, a pulsed release delivery system, and time-controlled drug release systems, have been developed using nanotechnology to design precision medicines. Further, 3D printing is also being investigated for the design of nanomedicines [39,40].

Thakkar et al. developed a tablet using the concept of fill density by using 3D printing technology. The solid dispersion and fabrication of the tablet by adding the pharmaceutically active agent of different characteristics, i.e., weakly acidic drug and less water-soluble drug into hydroxypropyl methylcellulose (HPMC) acetate succinate, was achieved by using a merger of hot-melt extruder and fused deposition modeling (FDM) based on 3D printing processes. The effect of fill density over the range of 20–80% concentration was also evaluated. A significant effect and a strong negative correlation were observed due to fill density during an in vitro dissolution study over the range of different pH. Increasing the concentration of infill up to 80% allowed the release of the active ingredient in a controlled fashion. This study suggested the useful role of 3D printing technology in designing the tablet for the purpose of a controlled-release delivery system [41].

Algahtani and coworkers developed a shell with the help of extrusion-assisted 3D printing technology to encapsulate the immediate-release tablet. The shell was constructed using cellulose acetate to modulate the immediate release of propranolol hydrochloride to sustained release. The shell was fabricated with the different portions of cellulose acetate (rate-controlling polymer) and D-mannitol to create pores in the shell during the dissolution process. The enclosure of the immediate release tablet in a 3D printed shell demonstrated sustained release behavior. In addition, the drug release, after enclosure with a 3D printed shell, followed Korsmeyer–Peppas kinetics and non-Fickian diffusion. Further, the release behavior of the 3D printed shell-enclosed tablet indicated applicability of the shell in the delivery of a Biopharmaceutics Classification System Class-I drug for the better adherence of patient-to-dosage regimen [42].

Zhang and coworkers produced strontium-incorporated and mechanically strong mesoporous bioactive glass scaffolds having specific architecture using a 3D printing technique. The scaffolds designed using 3D printing technique showed superior porosity and unvarying interconnected macropores. In addition, this 3D printed strontium incorporated mesoporous bioactive glass displayed approximately more than 170 times compressive strength in comparison with polyurethane-foam-template mesoporous bioactive glass scaffolds. Moreover, the 3D printed mesoporous scaffolds exhibited controlled and slow dissolution of strontium ions with superior bone regeneration activity and higher mechanical strength, which showed the importance of 3D printing technology in the production of bone-forming scaffolds [43]. Li and coworkers evaluated the feasibility of 3D printing technology to design a novel means for the controlled release of glipizide dispersed in the filaments of polyvinyl alcohol (PVA). A dual nozzle 3D printer was used to prepare the drug containing filaments of PVA. The drug release pattern was found to fit the Korsmeyar–Peppas release model. The study concluded that by using a double-chamber design and by varying the concentration and distribution of drug in the device, the release of glipizide could be achieved in controlled fashion [44].

Gioumouxouzisand coworkers formulated an oral device for the controlled release of hydrochlorothiazide, as a model drug. The orallycontrolled-release drug delivery device was prepared by dual extrusion–FDM (DE–FDM) 3D printing technique with water soluble mannitol and PVA. Characterization of the controlled device by various techniques confirmed that hydrochlorothiazide was incorporated in amorphous form to the dispersion. The drug release patterns in pH 1.2 and 6.8 confirmed the zero-order release behavior. Hence, it was suggested that 3D printing is useful in tuning the molecule for controlled-release purposes [45]. A controlled-release capsule-shaped tablet containing budesonide in the filaments of PVA was designed using FDM-based 3D printing. The release of budesonide was observed in the mid-small intestine, where sustained release behavior was depicted in the distal intestine and in the colon region. Therefore, the drug release pattern of budesonide from the caplets confirmed the potential role of 3D printing in the design of controlled-release formulations and precision medicines [46]. Zhang et al. aimed to achieve the zero-order release of acetaminophen by dispersing it into the matrix of hydroxypropyl methyl cellulose (HPMC), followed by production of tablets of different geometry with the help of 3D printing to ensure the controlled release of the drug. In addition, tablets of different inner core fill densities and with various external shell thicknesses were evaluated to ensure the controlled release of the drug. In this study, HPMC was found to be useful to achieve a steady and constant rate of drug release [47]. Kyobulaand coworkers used accurately controlled and solvent-free inkjet 3D printing to yield fenofibrate-loaded tablets with a honeycomb pattern. The tablet was prepared using FDA-approved and naturally occurring beeswax. The scientists speculated that the wettability and cell size of the honeycomb arrangement could be optimized to guarantee the personalized medicine to design tablets and medical devices [48]. Scientists are continuously working on different 3D printing technologies with nanotechnology to design controlled-release systems for development of precision medicines.

### 4.2. Nanofibers

Nanofibers are a new class of materials that are composed of fibers having a diameter in nanometer range, and exhibit high porosity and high surface-to-volume ratio. These fibers are generated from different polymers, including chitosan, cellulose, poly-ε-caprolactone (PCL), PLA, and copolymer of polylactic/glycolic acid (PLGA). The first ultrafine fibers were concocted and patented by Formhals in 1934 [49]. Since then, nanofibers have gained multifarious applications, including tissue engineering, wound healing, drug delivery, nanocomposites, filters, second-hand separator membranes, etc. Three-dimensionalprinting is the most widespread technique employing synthesizedelectrospunnanofibers, which include layer-by-layer fabrication of fibers. The setup comprises a printer that consists of numerous arrays of nozzles through which polymeric fluid is pushed. An electric field is used to draw the fluid out through the tiny fibers. One of its major uses in the field of tissue engineering is to synthesize 3D printed scaffolds. These scaffolds are porous and allow cells to permeate and, moreover, provide an ideal microenvironment for a myriad of proteins to be synthesized that aid in tissue repair.

Yu et al. concocted a 3D printed bone tissue engineering scaffold using infused PCL/gelatin-dispersed nanofibers into the meshes of a PCL printing scaffold. The porous structure of the scaffold plays an important role in eliciting cellular response [50]. Huang et al. synthesized dual-scale 3D printed scaffold for bone tissue repair using PCL nanofiber [51]. Ambrusand coworkers studied the effects of drug-loaded nanofibers on the low water solubility of the drug loratidine. Amorphous morphologies of electrospunnanofibers were fabricated using 3D printing electrospinning setup. The results depicted that the nanofibers exhibited a remarkable 26-fold increase in solubility and dissolution with 60% drug release from nanofibers, in comparison to only 4% release from the pure drug [52]. De Araujo et al. concocted PCL nanofilaments containing the bioactive ceramics nanohydroxyapatite (nHAP) and Laponite (Lap) via an extrusion process using a 3D pen. Nanohydroxyapatiteaids in osteogenesis and Lap is a synthetic smectite clay that promotes bone formation, cell proliferation, and attachment. The results suggested that these nanofilaments were nontoxic and further depicted that they presented a good dispersion of nHAP and Lap into polymeric matrices, and thus exhibited potential for bone tissue regeneration [53].

### 4.3. Thermoresponsive Hydrogels

Thermoresponsive hydrogels have been explored recently for 3D bioprinting application due to their reliable printing with prerequisite shape, and can be tuned easily to the sol-gel state by varying temperature due to which rapid gelation is achievable [54]. Some of them also provide good cell compatibility and suitable atmosphere for cell growth, so they are being explored widely in tissue engineering for creation of 3D structures that provide natural vascularization and microenvironment [55,56]. There are several thermoresponsive hydrogels with specific mechanical and biological characteristic and, depending on origin, polymer chain length structure, and gelation mechanism, they can be utilized for 3D printing. Some of the thermoresponsive hydrogels together with their applications are described in Table 2.

A hybrid bioprinting system is being investigated widely for vascularization and tissue engineering [57,58]. In a recent study, Kesti et al. designed a photocrosslinked PNIPAAm grafted hyaluronan (HA-PNIPAAm) thermoresponsive polymer, which confers fast gelation and immediate postprinting structural fidelity with methacrylatedhyaluronan, which safeguard long-term mechanical stability, and can be printed as high-resolution scaffolds with good feasibility. These two crosslinked polymers showed a lower critical solution temperature, 25.7–29.7 °C, and demonstrated rapid gelation upon interaction with a 37 °C-heated substrate, giving the 3D printed construct an instant structural fidelity. It was observed that concentration and charge of additional biopolymers and the cells presence influenced the gelation temperature of hydrogel and final storage modulus of the construct, but a highly printable system was obtained [59].

Fabrication of 3D tissues that mimics native organs using cell accumulation and 3D printing technique have been pursued globally in the fields of regenerative medicine and tissue engineering [60,61]. Tsukamoto et al.evaluated the 3D-shaping ability of a thermoresponsive polymer, hydroxybutyl chitosan, using a robotic-dispensing 3D printer, and constructed 3D tissues whose cell orientation and shape were controlled precisely to about 50 μm thickness using layer-by-layer (LbL) techniques [62]. In another study, Abbadessaand coworkers utilized a thermoresponsive hydrogel system based on methacrylated chondroitin sulfate and a thermosensitive poly(*N*-(2-hydroxypropyl) methacrylamide-mono/dilactate)-polyethylene glycol triblock copolymer (M15P10) to design a suitable biomaterial for scaffold manufacturing. The 3D printing of this hydrogel generated constructs with good handling properties and tailorable porosity. Finally, viability of embedded chondrogenic cells were found to be >98% and proliferating over a culture period of six days [63].

A novel thermoresponsive cell printing ink made up of aqueous polyurethane (PU) dispersion system was synthesized by Heishand coworkers to print neuro stem cells (NSCs). Printed NSCs had favorable differentiation and proliferation because the PU hydrogels showed appropriate modulus and chemistry (25–30% PU2). NSC-laden PU2 hydrogels promoted healing of a damaged centra nervous system (CNS) after injection in the zebrafish embryo neural injury model. In addition, when 3D-printed NSC-laden 25% PU2 constructs were implanted, the function of zebrafish with traumatic brain injury was saved. It was found that NSCs embedded in suitable PU hydrogels may have the potential to save the function of impaired nervous systems in neurodegenerative diseases [64]. Vascular engineering deals with the fabrication of 3D vascular constructs having proper blood circulation. In a recent study, a boronate ester hydrogel with 3D printing ability and self-healing ability was prepared by Tsai and coworkers as sacrificial materials to mould tubular microchannels in a nonsacrificial gel. The hydrogel was prepared via the copolymerization of PNIPAAm as thermoresponsive polymer, with PVA for gel formation and pentafluorophenyl acrylate for postmodification purpose. Additionally, cellulose nanofibrils were further added for facilitation in 3D printing of the hydrogel. Vascular endothelial cells seeded, proliferated, and attached in the microchannels of this construct. The results suggested that the glucosesensitive, 3D printable sacrificial hydrogel may be beneficial in the future for construction of easily removable 3D constructs to fabricate vascular-rich biomimetic structures [65].

Lin et al. created 3D printed bioceramics (3DP-BCs) osteoconductive scaffolds with a bicontinuous phase and microchannel pores based on the reverse negative thermoresponsive hydrogel, PNIPAAm-co-(methacrylic acid), by using robotic deposition additive manufacturing. The bone-healing analyses of 3DP-BCs scaffold in the case of calvarial bone defects in a rabbit model and cell viability assays depicted that 3DP-BC scaffolds have efficient new bone regeneration capacity with optimal continuous pores and suitable compressive strength, which provides an adequate microenvironment for cell growth and enables the protection of calvarial defects [66].

### 4.4. Hydrogels

Hydrogels are water-containing 3D networks with good structural integrity. Their characteristics include superabsorbent, flexible, highly stretchable, and self-healing properties, which have enabled them to be extensively used in different fields of science [67]. They exist in a colloidal state with water as a continuous phase [68]. However, the existing hydrogels have been found to be useful only for single purposes. Currently, work is still being done to make hydrogels with more than one property in order to expand their application [69]. Different bioprinting technologies such as inkjet printing, microvalve-based printing, extrusion-based printing, laser-assisted forward transfer, and SLA have brought a revolution to the field of 3D printing technology of hydrogels. Extrusion-based printing has been the most widely explored technology over the last decade [70]. Some of the current research work done in the field of hydrogels by using 3D printing technology to explore more of their advanced utility are discussed below.

Ma et al. formulated hydrogels of cellulose nanocrystal with viscoelastic properties by using an extrusion-based 3D printing technology. Varying concentrations of cellulose nanocrystal (CNC) in the range of 0.5–25 wt.% were used to make hydrogels in order to check their rheological properties and printability. CNC hydrogels at 20 wt.% showed the best optimal print resolution and fidelity, with a high degree of orientation (72–73%) of CNC alignment along the printing direction [71]. Abouzeid et al. manufactured aqueous hydrogels for 3D printing by directly mixing PVA cellulose nanofibers with sodium alginate and hydroxyapatite (HAP), and have been proven to be excellent for bone tissue engineering [72]. Semisolid tablets of theophylline with varying drug loading in the range of 75–125 mg was manufactured in association with HPMC K4M or E4M hydrogels by Cheng and coworkers, whoused the technique of extrusion-based semisolid 3D printing. The hydrogel with high excipient concentration was found to have relatively high yield stress, storage modulus, and hardness [73].

Extrusion-based hybrid hydrogels of gelatine and alginate (G/A), with different total solid concentrations (3%, 5%, and 7%) and G/A ratios (1:2, 1:1, and 2:1) were prepared, which have porous structures with the potential to encapsulate and deliver bioactive compounds such as enzymes, vitamins, antioxidants, and probiotics [74]. Zhang et al. used the phenomenon of electrostatic interaction and hydrogen bonding to physically crosslink poly (sulfobetaine-co-acrylic acid)/chitosan–citrate to form a double network hydrogel. The hydrogel thus prepared showed high transparency, excellent self-healing properties (as high as 95.4%), good electrical conductivity (conductivity 0.11 S/m), and reasonable sensitivity [75].

### 4.5. Nanocapsules

Nanocapsules are spherical colloidal structures having a hollow core surrounded by a polymeric shell that can load both hydrophilic and hydrophobic drugs. The size range of these nanocapsulesvaries from 10 to 1000 nm. They have advantages over other delivery systems by acting as a smart carrier with the ability to increase bioavailability, efficacy, and safety [76]. Antiwrinkle lotion having vitamin E nanocapsules was one among the first products available in the market [77]. For designing of many nanoformulations, 3D printings arenow used to decrease the time taken during manufacturing, to decrease wastage, and to increase automation with the help of many CADs. Different techniques, such as SLA, FDM, selective laser sintering, liquid inkjet printing, pressure-assisted microsyringes, have been used for 3D printing of nanoformulationssuch as nanocapsules.

Beck et al.initially developed drug delivery devices by combining 3D printing with nanotechnology to deliver deflazacort in the form of 3D printed tablets loaded with nanocapsules, which were designed using FDM. They soaked the 3D printed filaments with different polymers (PCL, Eudragit^®^ RL100, with or without mannitol as a channeling agent) and different infill percentages in the suspension having deflazacortnanocapsules. The nanocapsulesprepared showed a positive zeta potential of +6.87 mV and a size range of less than 0.284 µm, with a very narrow polydispersity index of 0.10 and an in vitro drug release of 65% after 24 h. Further, they observed that there was an increase in the drug loading (deflazacortnanocapsules) into 3D printed filament with increase in soaking time, drug release from 3D printed tablets by Fickian diffusion, and case-2 transport [78]. Rupp et al. prepared small core shell capsules by a dual-step 3D printing design using a CAD program, in which top and bottom layers were loaded tightly with thermoplastic polymer PCL and nanocapsules using FDM. Middle layers were loaded with oil (linalool, limonene, trivalent alkyne, and farnesol) by inkjet print head, forming microsized capsules having a size range of 200–800 µm and containing nanocapsules. These capsules, designed by using 3D printing, showed thermal stability up to a temperature of 80 °C [79].

### 4.6. Nanoparticles

SLA is a laser-based 3D printing system that uses anLbL assembly method to create aligned micro- and macrosize 3D constructs, but is not capable of achieving nanoscale architecture. Lee et al. integrated core shell nanoparticles onto the nerve scaffolds printed using SLA, thus successfully attaining nanotopology and a sustained delivery of bioactive molecule for enhancing nerve regeneration in peripheral nerve injuries [80]. Three-dimensional printing can also be used to produce biomaterials with customized scaffolds and predetermined architecture, shape, porosity, pore size, and pore interconnectivity, with high reproducibility. Nanoparticles are, however, used to enhance the surface and biological characteristics of such scaffolds.

Roh et al. added magnesium oxide nanoparticles to PCL and HAP composites for improving bone regeneration. Magnesium, an essential natural mineral for bone growth and also a biodegradable metal, proved to be effective in promoting osteoblast cell proliferation and the 3D composite scaffolds of PCL/HAP/magnesium oxide nanoparticles with interconnected pores, which showed maximum bioactivity compared to other groups [81]. In another attempt to explore the role of nanoparticles and 3D printing technology in bone regeneration applications, Abdal-hay et al. blended bioresorbable magnesium hydroxide nanoparticles with the degradable polymer PCL and manufactured the composite using 3D printing technology. Unlike polymer-only scaffolds, the incorporated nanoparticles promoted osteoblast metabolic activity, attachment, and proliferation [82].

From all these research reports, we can summarize that 3D printing is being investigated with promising results for the design of precision medicine and nanomedicine, where 3D printing gives the opportunity to design the scaffold/device/system with fine tuning to match individual needs.

## 5. Applications of 3D Printing-Based Nanomedicines

Three-dimensional printing-based drug delivery systems offer limitless advantages such as personalized drug delivery systems—controlled delivery that can be tailored for individual specific purposes. Besides all these, 3D printing involves simple techniques for cost-effective production that is easily scalable. A summary of different 3D printing technologies used in additive manufacturing of nanomedicines is given in Table 3. Currently, healthcare systemsare employing 3D printing in preclinical and clinical setup for personalized dosage forms—for example, tablet, capsule, dentistry, tissue/organ regeneration. This section discusses the existing medical application of 3D printing with potential for translational purposes.

### 5.1. Implants

Tailor-made or customized implant designing can be done by employing the techniques of 3D printing. Reasons that attenuate the need for customized implants are (i) patients deviating from standard size, as noted, and (ii) improvement in fitting and movement of individuals. Further, biocompatible materials involved for development of implants and devices aids another advantage of wide acceptance in targeted drug delivery. Brief descriptions of implant devices designed using 3D printing are given in Table 4 [95,96,97].

Microswimmer devices are fabricated using 3D printing technology for various applications, including nanomedicine and precision medicine, as described in Section 2 of this review. Microswimmer device mechanism depends on three stages: i.e., loading, transportation, and release. The mechanism for transportation focuses on the concept of engineering to control its movement to target tissues and organs. Microswimmers, on excitation or sensation of a stimulus, move in desired direction by focusing on shape, and make their way through in vivo vascular systems. Amongst all methods used to engineer these devices, magnetic actuation is the most prominent because the magnetic fields are noninvasive [21,95]. Microimplants are a high-precision targeted device being investigated for tissue regeneration and tissue function restoration [95]. Zhu and coworkers designed a scaffold manufactured using a 3D printing material extrusion method, which exhibits a prolonged release profile [96]. Wu and coworkers incorporated many layers of different drugs in a multidrug implant, manufactured by 3D printing technology binder jetting, to obtain dual-pulsed release for treatment of bone tuberculosis [97].

Three-dimensional printing technology is being explored in spine surgery, particularly in vertebral skeleton model printing. Due to simplicity and relatively low cost, it is being favored by both doctors and patients. Further, 3D printing is also being explored to design and prepare a navigation template that can guide an internal fixation screw to obtain to guide its movement. Applications of 3D printing are being explored in manufacturing of medical devices as it enables personalization of implants and medical devices and can ensure negligibly invasive diagnosis [98,99]. Undoubtedly, clinical application of 3D printing is enormous, and it can be explored successfully in the design of personalized implants and medical devices, as suggested by preliminary research.

### 5.2. Anticancer Drug Delivery

Anticancer drugs face difficulty in reaching the site of action. Their accumulation may cause severe toxicity in various noncancerous organs. Use of conventional systems such as oral dosage forms or intravenous (i.v.) injection also possesses drawbacks because of drug-poor solubility, which may lead to huge suffering in the case of cancer patients. Scaffolds have been prepared by 3D printing technology employing polymers such as PCL and PLGA for delivery of anticancer and antibiotics. These have also been developed as patches with proper geometry and showed drug release for four weeks with well-defined release kinetics, which could enhance patients’ acceptability [5,100,101,102].

Maher et al. designed titanium implants using 3D printing technology to fabricate unique micro- (particles) and nanosurface (tubular arrays) topography for osseointegration and localized delivery of the anticancer drug, doxorubicin and apoptosis-inducing ligand (Apo2L/TRAIL). These 3D nanosurfaces could be explored for localized chemotherapy of primary and secondary bone cancers together with fracture support [103]. Chen and coworkers fabricated a 3D printed microfluidic chip for delivery of combinational chemotherapeutics. These chips contain multichannel helical structure for mixing the chemotherapeutic solutions rapidly.The combined effect of the anticancer compounds showed the synergistic cytotoxic effect on A459 cells [104]. In another work, 3D printed microneedles were prepared by SLA using biodegradable resin. These microneedles were used in in vivo tumor regression studies and showed significant reduction in tumor weight [105]. Three-dimensional printed calcium phosphate podiums with premeditated porosity were designed for encapsulation of curcumin in a liposome. Curcumin released from the 3D printing scaffold showed significant cytotoxicity toward in vitro osteosarcoma (bone cancer) cells, whereas it promoted osteoblast cell (healthy bone cell) viability and proliferation [106]. The treatment of a fatal and complex disease such as cancer requires designing a smartly engineered and personalized therapeutic system, which could be made possible in the near future by using 3D printing and nanotechnology.

### 5.3. Wound Healing

There is huge demand existing for personalized innovative materials for developing additively manufactured structures for wound healing. Nanotechnology-based approaches have shown potential to resolve many challenges associated with current medicine systems, but their safety still remains a major question. Although such approaches have investigated antibacterial nanoparticles as carriers of active substances that improve wound healing, they are nevertheless difficult to produce at industrial scale. There werealso antimicrobial patient-specific wound dressings manufactured from PCL by incorporating zinc, silver, and copper. Metal-loaded filaments wereobtained by using hot-melt extrusion, and nose and ear 3D models wereprinted. Wound dressings also showed extended release of various metals and bactericidal features.

The 3D printed hybrid scaffolds, which were focused on homogenized pericardium matrix and poly (ethylene glycol) (PEG), were made to promote healing of wounds in the case of vascular grafts supporting injured vessel replacement after surgical reconstruction. Minimization of inflammatory signal from macrophages was observed on incorporating homogenized pericardium into PEG matrix, which can affect the scaffold modulus [107]. Scientists have incorporated functionalized nanosized cellulose for fabrication of 3D bioplotters with tailored tissue, engineered at centimeter scale and optimized with alginate/methylcellulose hydrogel. This could be used for regenerative therapy in cartilage and fats with an additional option to overcome tissue defects [108]. Lee and coworkers successfully demonstrated 3D printing of human skin using LbL technology. Three-dimensionalnanobioprinting provided a specific advantage of resemblance with human skin, which simulates shape, physical dimensions, architecture, and structure [109]. A three-dimensional printed composite of zinc oxide (ZnO) nanoparticles with photocatalytic activities was encapsulated within hydrogel (alginate) constructs, for antibacterial activity to assist in wound healing. Three-dimensional printed constructs revealed significantly greater pore sizes and enhanced structural stability compared to manually casted samples. Bacterial resistance testing on *Staphylococcus epidermidis* indicated that the addition of ZnO nanoparticles to the gels decreased bacterial growth when compared to the blank alginate gels. Cell viability of STO-fibroblast cells was not adversely affected by the addition of ZnO nanoparticles to the alginate gels [110].

### 5.4. Tissue Engineering and Regeneration

Tissue engineering plays an important role in biocompatible implant fabrication for replacing nonfunctional or damaged tissues. This field takes biocompatible materials, growth factors, and live cells to fabricate implants for helping in normal growth of tissues. The additive manufacturing allows the fabrication of various 3D printed models that later mimic the microscopic network of connective tissues. There are various merits of 3D printing in the field of tissue engineering, one of them lies in its ability to provide geometric complex structures by accurately placing the materials in 3D space (Figure 5).

Additive manufacturing offers a better tool for the manufacture of implants helping in bone regeneration. Managing critical-size bone defects remains a challenge. There may not be a guarantee of bone integration where there is filling of larger bone defects that are required with bone grafts. The present surgical management option includes bone graft vascularization, which is unfavorable and highly technical, or to perform “Masquelet procedure”, which requires many operations and can maximize the morbidity. Additive manufacturing provides potential implant solutions promoting bone regeneration and vascularization. The porosity is an important feature for promoting bone in-growth. Three-dimensional printing provides a platform for creating high resolution and porous scaffolds from many materials such as metals and ceramics [112]. Lee and coworkers described printing and implantation of acellular HAP/PCL scaffolds in rabbits, which were doped with the transforming growth factor β3 (TGF β3). After application, it exhibited a full articular surface regrowth of proximal humeral joints [113]. Chang and coworkers demonstrated that a FDM-printed PCL trachea, which was coated with fibrin and mesenchymal stem cells (MSCs), exhibited neocartilage production and integration with native tracheal tissue [114]. Scientists are exploring 3D printing to obtain a scaffold of desired nanotopographyin polyvinylpyrrolidone, viaa printing suspension of calcium phosphosilicatenanoparticles having sizes rangingfrom 20 to 100 nm. Application of 3D printing in the design of nanomedicines could help in creating nano- and microscale scaffold with desired effects such as osseointegration, as required in bone tissue engineering, and can enable the design of geometry of scaffold matching with personalized geometry of defects [115]. We can conclude that combined application of 3D printing with nanotechnology can result in an outstanding outcome in the fabrication of nanomedicine and precision medicine.

### 5.5. Implication in COVID-19

Recently, global uncertainty arose due to pandemic COVID-19, which created a worldwide crisis. COVID-19 resulted in challenges and disruptions in supplies of various kinds of medical emergency equipment, due to the shutdown of logistics and transportation. During this period, 3D technology potential was harnessed to provide the needed medical and testing devices, personal protective equipment (PPE), emergency dwellings, visualization aids, and personal accessories. Three-dimensional technology was also used for training healthcare workers with a visualization video explaining the method of using PPE kits and other medical devices [31,32,33,34]. Table 1 presents the description of masks/PPE kits designed and fabricated using 3D printing for protection from COVID-19. Erickson et al. explored an innovative solution by creating an adaptor for the Flyte helmet, so that they could be converted to PPE [32]. Replication of body organ parts accurately was possible by 3D printing of life-size medical dummies used to train the medical staff for COVID-19 swab testing. Three-dimensional printing could be used for the development of nasopharyngeal swabs and to meet the requirement of surplus demand of swabs [116].

### 5.6. Peptide Delivery

Microneedles, which possess micron-sized needle arrays on matrix surface, enhance skin penetration of bioactive molecules. All these microneedles can be more effective to deliver many macromolecules through skin rather than a traditional patch. Recent advancements in higher resolution 3D printing methods, which fabricates small structures, thus provides huge applications of this printing in manufacturing of microneedles. While conventional techniques of microfabrication are restricted only to microneedles having simple geometry, the newer 3D printing technologies enable microneedle fabrication that possesses more complex and composite geometry [117].

Pere and coworkers prepared polymeric microneedle patches of a biocompatible nature by employing the SLA method to deliver insulin through skin. The biocompatible resins were photopolymerized to prepare microneedles of cone and pyramid geometry, which was then followed by inkjet (IJ) print coating of insulin formulations. The fabricated microneedles possessed a better piercing capacity and mechanical strength. They released insulin very rapidly within a span of 30 min irrespective of the shape of the microneedle [118]. Employing micro-SLA, Lu and coworkers prepared microneedles of poly (propylene fumarate) by incorporating anticancer drugs for treating skin carcinoma. For improving mechanical strength and adjusting viscosity, they mixed diethyl fumarate with poly (propylene fumarate). Their microneedles released the anticancer drug, dacarbarzine, for period of five weeks [119].

Ultrashort peptides might be employed for the purpose of bioprinting applications as bioinks. The ultrashort peptide moieties have the property to assemble themselves into hydrogel having nanofibrous topographic structure, which closely resembles collagen and hence mimics tissue extracellular matrix native architecture. Considering an example, ultrashort peptide hydrogel exhibited biocompatibility by maintaining stem cell 3D culture and intestinal epithelial cells (Caco2) organotypic culture. There was retention of embryonic stem cell potency, which were encapsulated in ultrashort peptide hydrogels, when Tra-I-60, Tra-I-81, Oct4, and Nanog were used as markers of pluripotency. There was differentiation of human mesenchymal stem cells thatwas present in the peptide hydrogels into adipogenic lineage on exposure to a defined culture condition. The peptide hydrogels might also offer appropriate nanotopographyand a 3D microenvironment to support primary cell organotypic culture along with stemcell 3D culture [120].

### 5.7. Detection and Diagnosis

During the last few decades, medical imaging has been reported as one of the emerging techniques used in the healthcare sector. Three-dimensional visualization of internal organ and tissues with proper image navigation provides a pivotal edge to clinicians in diagnosing the diseases. Three-dimensionalmedical technology is done in three steps: image acquisition, image postprocessing, and 3D printing. Recently, additive manufacturing came in to the picture for tremendous progress in diagnostics, which involves CAD having an ability to be converted to 3D printed pieces [121]. Additive manufacturing techniques are cost effective, thus applicable in fabrication of analytical equipment and tools for diagnostics. Three-dimensional printed microfluidic devices are the most captivating application of 3D printing in diagnostics. This technique is now employed for automatic andmanual-type diagnostic strategies. Prominent advantages are offered by microfluidic devices due to their ability to manage small volumes [87,122].

Evidence confirms that 3D printing can form assisting devices for already available diagnostics such as polymerase chain reaction (PCR), and helps in remote sensing through mobile phones [123,124,125]. Diagnostic devices accompanied with electronics and numerous circuits can also be manufactured by 3D printing. Another novel modified version of 3D printing is 3D bioprinting, which involves encapsulation of biostructures and chemicals such as cells, proteins, and tissues in printable bioink solutions [126]. Enzyme-linked immunosorbent assay (ELISA) is one of the most commonly used assay techniques for diagnosis of infectious diseases such as acquired immunodeficiency syndrome (AIDS); it is based on the principle of antigen–antibody interaction. Three-dimensional printing platforms can enhance the effectiveness of the former because ”3D well” has a larger surface area than a conventional titer plate and renders good sensitivity; hence, diagnosis can be performed easily and rapidly [127].

## 6. Limitations and Constraint

Every manufacturing process encounters both advantages and limitations, which is similar and true with 3D printing. Even 3D printing is gaining ground day-by-day, but when it comes to regular practice in pharmaceutical production, there is long way to go. Three-dimensional printing still possesses certain limitations to technical issues, along with legal and regulatory concerns. Production yield of 3D-enabled technology in comparison to conventional manufacturing technology is very low [2,128]. The foremost example is that the conventional method with a single pressproduces around 15,000 tablets in one minute, while for 3D printing it takes from 2 min to 2 h for one tablet. Limitation of yield hinders pharmaceutical production, but experts argue that the advantage with 3D printing is that it can beestablished in a small set-up, which might be at pharmacy shops or hospitals. Another technical issue arises; the thickness of films finally formulated depends on the choice and source of material and the presence of impurities.

Other specific challenges can be attributed to the production method for 3D printing. While working with a laser-dependent method, it is important to have well-established material that provides a customized release profile; hence, it is necessary to previously study the degradation potential of drugs and polymers. A formulation scientist tried to develop ramipril3D-immediate-release tablet, but when the drug was melted, it generated impurities and formulated a highly unstable product. After further investigation and proper knowledge about the excipient drug properties, however, a stable product was formulated with desired release profile.

Consideration of regulatory concerns is important in development of this technology. By comparing the in vitro–in vivo correlation (IVIVC), one can establish its product in the market, but it could be very challenging for the manufacturer to get regulatory approval of a formulated 3D-based product because no specific guidelines were available, previously. As the technology accelerates its development and establishment, more questions pertaining to its release profile arise. For example, if all the materials of dosage form release at one time, then it may cause toxicity; or if a patient is undergoing any other treatment or medication, then delivering a 3D printed dosage form may cause other physical or chemical interactions. Another challenge is market access for this technology. Can small hospitals and pharmacy shops establish the 3D technology at small scale and maintain the quality standards as established by innovator? These questions must be answered before 3D technology will be established globally as a foremost tailor-made technology. Although 3D printing plays a beneficial role in personalized medicine, it still possesses a few issues related to safety, monitoring, trust, and regulation. If the 3D manufacturing process fails, this will lead to risk for individuals; thus, keen monitoring is essential at regulatory level.

## 7. Regulatory Aspects

Tremendous use of additive manufacturing or 3D printing technology is being noticed. One of the specific features of 3D printing is that any devices with any kind of simple and complex geometry can be produced easily to meet the required criteria. Hence, to regulate the use of 3D printing in the field of manufacturing devices, the FDA issued a draft in May 2016 entitled “Technical Considerations for Additive Manufactured Devices”. Three-dimensional printing is also useful in the production of special devices that are suitable to unique anatomy of the individual patient. In this guideline, one has to clearly explain each step of manufacturing the 3D product. This guideline also includes recommendations for explanation related to how each parameter affects the property of product, and why this method is employed for production [129].

FDA guidelines also mention clearly that any material used for the additive manufacturing must come from an identifiable source. Similarly, other possible sources for the material should be identified. Any property of material that should affect the interlayer bonding must be the representative feature of the final finished device. FDA also describes the issue regarding software that is used for the design and production of devices by 3D printing. As the FDA allows the marketing of various 3D printed devices for biomedical application, the prerequisites for the material used in the 3D printed product are being cleared with passage of time.

## 8. Conclusions and Summary

This manuscript conceptualized and highlighted the applications of 3D printing-based technology in the design of nanomedicines and advancement in novel drug delivery systems. Further, we also reviewed the 3D-based nanomaterials and their versatility in personalized medication, with consideration of regulatory perspectives.

Currently, the 3D-based approach as drug delivery technology is in infancy, but in the near future it may emerge as a promising technology and transform at commercial level to achieve tailor-made release profiles. The goal of advanced medication or therapeutics is to serve a current patient’s needs. The goal of this manuscript is to provide a clear presentation and describe3D-based technology state-of-the-art. A wide range of nano-based biomaterials is available for processing 3D-based medication and assuring its compatibility. As discussed, 3D printing has envisaged its potential in tissue engineering, wound healing, target-based delivery, anticancer delivery, and others. All of the applications discussedare constrained and limited to preclinical application, and only due to limited regulatory guidelines and well-established biocompatibility studies in humans.

## Figures and Tables

**Figure 1 nanomaterials-11-00420-f001:**
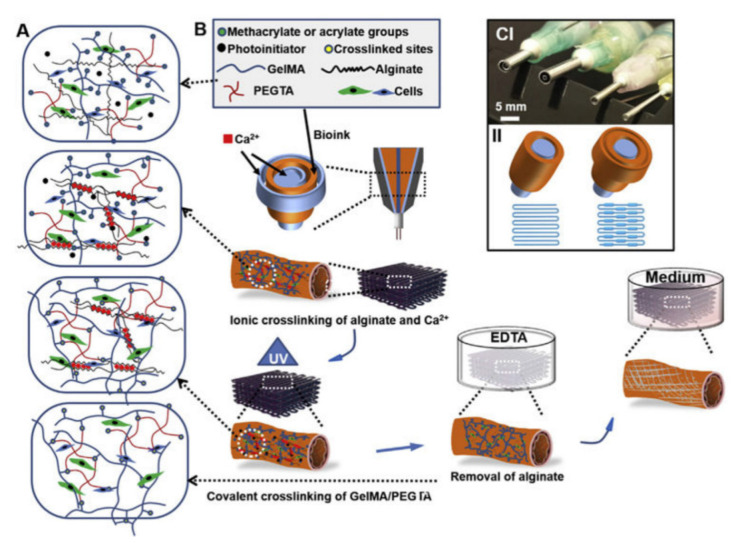
Three-dimensional printing-based design of perfusable vascular construct using crosslinkedbioink. (**A**) Graphical presentation of crosslinking processes of the bioinks, where alginate, GelMA, and PEGTA are ionically and covalently crosslinked, respectively, upon exposure to CaCl_2_ solution and UV light. (**B**) Schematic representation of the procedure for bioprintingperfusable hollow tubes with the cell-encapsulating blend bioink and subsequent vascular formation. (**C**) The designed multilayered coaxial nozzles and schematic diagram showing fabrication of perfusable hollow tubes with constant diameters and changeable sizes. (GelMA: Gelatinmethacryloyl; PEGTA: 4-arm poly(-ethylene glycol)-tetra-acrylate].) (Reproduced with permission from [10]).

**Figure 2 nanomaterials-11-00420-f002:**
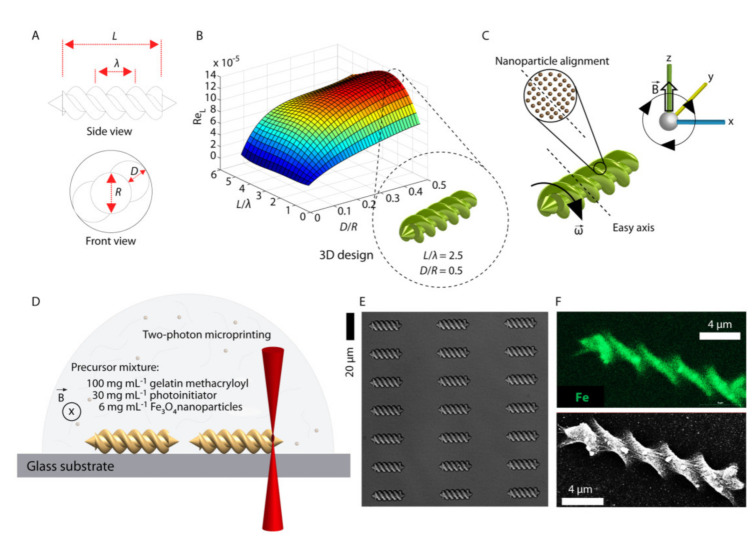
Diagrammatic representation of designing and 3D printing-based fabrication of biodegradable hydrogel microswimmers. (**A**) Empirical design of the double-helical microswimmer. (**B**) Computational fluid dynamics simulation for Reynolds number with respect to L/λ and D/R ratios, calculated for water at room temperature. The maximum forward swimming velocity was found with L/λ = 2.5 and D/R = 0.5 for the given design-space sweep study. (**C**) Alignment of the magnetic nanoparticles that defines an easy axis normal to the helical axis, thereby allowing rotational motion under rotating magnetic fields. (**D**) Three-dimensionalfabrication of the microswimmers using two-photon polymerization. During the fabrication process, a continuous magnetic field was applied to keep the nanoparticles aligned. (**E**) Optical microscope differential interference contrast (DIC) image of a microswimmer array. (**F**) Energy-dispersive X-ray spectroscopy mapping of iron, confirming the homogeneous embedding of the iron oxide magnetic nanoparticles inside the microswimmer body. (Reproduced with permission from [21]).

**Figure 3 nanomaterials-11-00420-f003:**
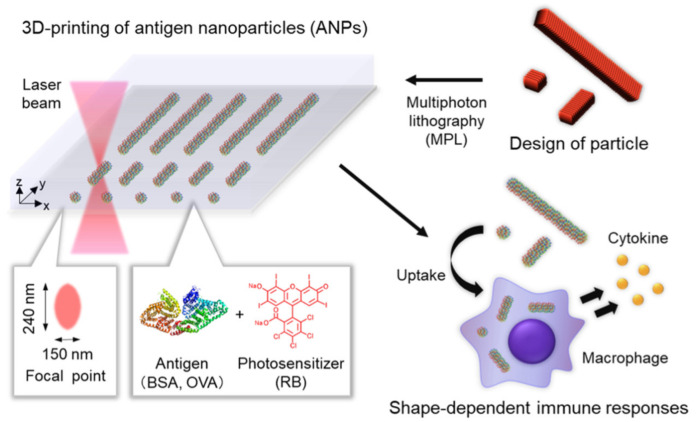
Graphical representation of designing ANPs using 3Dprinting where a photoresist composed of an antigen and a photosensitizer were used to fabricate shape-defined ANPs using multiphoton lithography. The ANPs designed could be taken up by macrophage and trigger adaptive immune responses depending on particle shape. (Reproduced with permission from [22]).

**Figure 4 nanomaterials-11-00420-f004:**
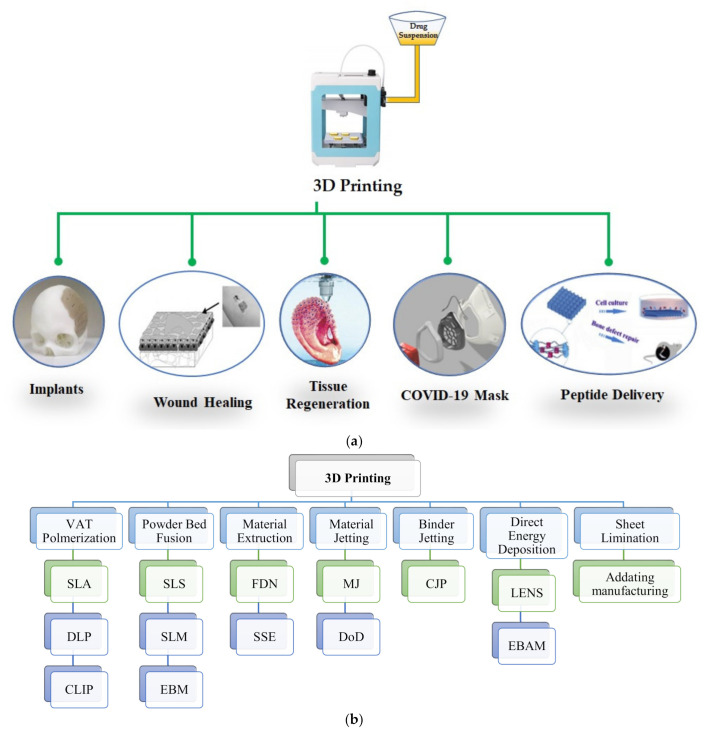
(**a**) Diagrammatic representation of applications of 3D printing in personalized nanomedicines. (**b**) Schematic presentation of different approaches of 3D printing in personalized nanomedicines: CLIP-continuous liquid interface production, CJP—color jet printing, DLP—digital light processing, DoD—drop-on-demand, EBAM—electron beam additive manufacture, EBM—electron beam melting, FDM—fused deposition modeling, LENS—laser engineered net shape, MJ—material jetting, SLA—stereolithography, SLM—selective laser melting, SLS—selective laser sintering, and SSE—semisolid extrusion. Redrawn with permission from [38]).

**Figure 5 nanomaterials-11-00420-f005:**
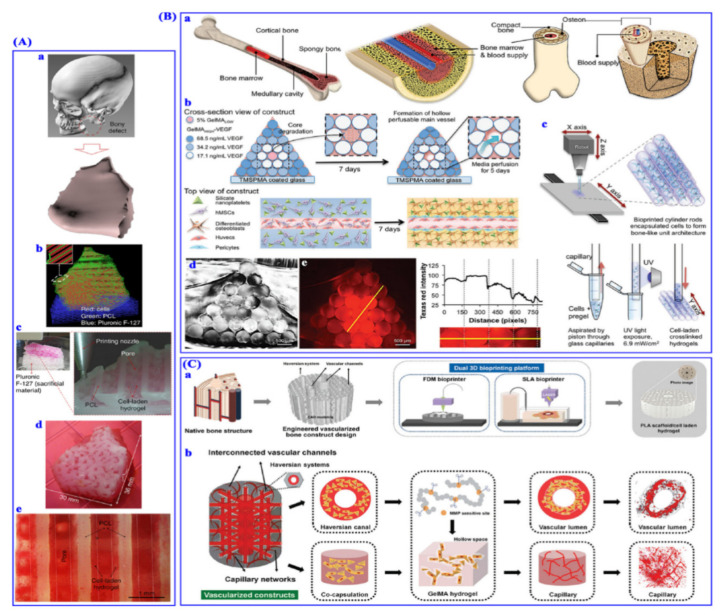
Three-dimensional bioprinting for bone regeneration. (**A**) Reconstruction of mandible bone defect. (**a**) Three-dimensional CAD model from human CT image data. (**b**) Visualized motion program to establish 3D defect architecture. (**c**) 3D bioprinting process with the integrated tissue–organ printer(ITOP) system. (**d**) Picture of the 3D bioprinted defect construct. (**e**) Calcium deposition showing osteogenic differentiation in the printed bioconstruct. (**B**) Biofabrication of osteogenic and vasculogenic 3D bone architectures. (**a**) Schematic of native bony structure. (**b**) Schematic of the bone bioprinting strategy. (**c**) Schematic of 3D bioprinting individual cell-laden cylinders. (**d**) Cross-section of the bioprinted pyramidal architectures. (**e**) Conjugation of gradient Texas Red to −COOH-modified GelMA. (**C**) Biofabrication of vascularized bony biphasic constructs. (**a**) Schematic of dual 3D bioprinting to engineer the vascularized bony constructs. (**b**) Design schematic of the vascularized bone constructs. Reproduced with permission from [111]).

**Table 1 nanomaterials-11-00420-t001:** Masks and other personal protective equipment (PPE) kits devised by using 3D printing technology for protection from COVID-19.

Face Masks	Description	References
Copper 3D NanoHack mask	Polylactic acid (PLA) filament was used as a flat piece, and then hand processed into its final 3D configuration after heating at 55–60 °C via forced hot air or by submerging it in hot water for the 3D-printed COVID-19 mask. The mask comprises a simple air intake hole containing two reusable filters with a screw to clasp the filters in place. The process showed the limitation with community-generated designs and requires improvement based on local testing and available technical base since this must be manually sealed to an airtight container.	[31]
The High Efficiency Particulate Air (HEPA) mask	A desktop 3D printer was used for designing and manufacturing a HEPA mask. PLA filament were best suited due to fitting the mask to the specific user after heat exposure, and also ensures the best possible air seal in open environments. The HEPA mask is fit in both male and female with space for a transferrable HEPA filter insert within a hole at the front side of the mask.	
The Lowell Makes mask	The Lowell Makes mask offers the advantage of printing without support and interchangeable front filter design. The mask uses foam padding on the inner side. The padding material is reusable and improves individual comfort, but the selection of sterilization methods must be considered cautiously.	
Hospital respiratory support apparatus	During the outbreak of COVID-19 in Italy, there was massive scarcity of masks. The Venturi-based valves are crucial components of that respiratory support equipment, which were difficult to reproduce. The pandemic situation necessitated the use of 3D printing to making valves for ease to local supply. The nonadjustable Venturi valves were mostly accessible by “GrabCAD” users. Therefore, all the designs of valves using 3D printing achieve fraction of inspired oxygen (FiO_2_) levels at oxygen supply supplemental rate. There may be a requirement of printing technology for ensuring airtight packs when model porosity alters FiO_2_ levels.	
3D printed arthroplasty helmets repurposed as PPE	A novel solution was proposed with adaptation of the standard helmet system given the shortage of powered air-purifying respirators and the potential shortage of N95 masks. The concept developed was to use a 3D printing approach to create an adaptor for the Flyte helmet to allow conversion to PPE.	[32]
3D printed mask adaptor	3D printed mask adaptor, outfitted with a sectioned portion of an N95 respirator, that maintains the N95 filtration standard and thereby multiplies the available number of masks. Maintaining the N95 standard requires a novel mask adaptor design that conforms to the face and seals around each component of the mask and filter.	[33]
3D printed masks for a Level-I orthopedic trauma	Utilization of 3D-printing capabilities to make 3D-printed face masks for orthopedic trauma provides the filters that were nearly equivalent to the filtration material that is found in N95 masks.	[34]

**Table 2 nanomaterials-11-00420-t002:** Thermoresponsive gels in 3D printing.

Thermoresponsive Hydrogel	Gelation Mechanism	Gelation Temperature	Concentration Used in 3D Printing (%w/v)	Technique	Cell Type	% Cell Viability	Application	References
Collagen	Crosslinking	~37 °C	0.1–3%	Pneumatic dispensing system	Keratinocytes	>94% (day 1)	Cartilage tissue engineering	[57]
p(HPMAm-lactate)-PEG Hydrogel	Thermal gelation and photopolymerizatin	21–40 °C	25–35%	Pneumatic dispensing	Chondrocytes	~85% (day 3)	Cartilagetissue engineering	[57,58]
Amino-terminated PNIPAAm hydrogel	Thermal and phototriggered tandem gelation	Room temperature	10–20%	Pneumatic extrusion	Chondrocytes	~98% (day 7)	Cartilage tissue engineering	[59]
5–8% Methacrylatedgelatin (GelMA)/gelatin	Thermal gelation followed by photocrosslinking	15–25 °C	5 and 8%	3D bioink printing system	Bone marrow stem cells (BMSCs)	92.9 ± 2.6 (day 7)	Tissue engineered scaffold	[60]
Agarose	Photopolymerizedgelation	<32 °C	1–5%	Pneumatic and mechanic extrusion	HUVECs	>70% (day 7)	Vascularization engineering	[61]

**Table 3 nanomaterials-11-00420-t003:** Summary of 3D printing technologies being investigated for pharmaceutical formulation development.

Printing Technology	Dosage Forms/System	Model Drug Used	Reference
Fused filament 3D printing	Tablet	Fluorescein	[83]
FDM 3D printing technology with HME and fluid-bed coating	Tablet	Budesonide	[46]
3D extrusion printing	Multiactive solid dosage forms (polyphill)	Aspirin, Hydrochlorothiazide, Pravastatin, Atenolol, and Ramipril	[84]
FDM 3D printing	Extended-release tablet	Prednisolone	[85]
3D printing	Tablet implant	Isoniazid	[86]
Desktop 3D printing	Bilayer matrix	Guaifenesin	[87]
FDM 3D printing	Modified release drug loaded tablets	5-aminosalicylic acid and 4-aminosalicylic acid	[88]
Extrusion-based printer	Multiactive tablet	Captopril, Nifedipine, and Glipizide	[89]
3D printer	Biodegradable patch	5-flurouracil	[90]
3D scanning and 3D printing	Antiacne filament	Salicylic acid	[91]
FDM 3D printing	Hot-melt extruded filaments	Acetaminophen and furosemide	[92]
Electrohydrodynamic printing	Patterned microscaled structure	Tetracycline hydrochloride	[93]
SLA 3D printing	Modified release tablet	4-aminosalicylic acid and Paracetamol	[94]

FDM—fused deposition modeling; HME—hot-melt extrusion; SLA—stereolithograpy.

**Table 4 nanomaterials-11-00420-t004:** Three-dimensionalprinting devices for targeted drug delivery.

Devices	Description	References
Microswimmer Devices	Microswimmer device mechanism depends on three stages, i.e., loading, transportation, and release. At each stage, microscale geometry contributes in a significant manner. Loading can be done by surface chemistry, passive adsorption, or mechanical trapping within arms or syringes. Mechanism for transportation focuses on the concept of engineering for motion to targeted tissues or organs. Stimuli responses are thermal, magnetic, and chemical on which release and transportation behavior depends. Microswimmers, on excitation or sensation of a stimulus, move in desired direction by focusing on shape, and make their way through in vivo vascular systems. Amongst all methods, magnetic actuation is prominent, as the magnetic fields are noninvasive. Microswimmer devices employ mechanisms that are either plated with magnetic material or have magnetic material incorporated, so would respond to a magnetic gradient or a rotating magnetic field.	[95]
Microimplants	Microimplants are high-precision targeting device with promising tissue regeneration and tissue function restoration. Manufacturing methods for microimplants are material extrusion methods (e.g., semisolid extrusion and FDM), selective laser sinistering, and binder jetting.	
Experimental studies on microimplants	Both Wu and coworkers and Zhu demonstrate incorporation of many drugs with programmable drug release profiles. Zhu and coworkers demonstrated a prolonged release profile from scaffold printing, manufactured by using a material extrusion method. Wu and coworkers incorporated many layers of different drugs, manufactured by binder jetting with an aim of dual-pulsed release.	[96,97]

## Data Availability

Not applicable.

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
