# Peer review of "3D Printing in Development of Nanomedicines"

_nanomaterials, 2021, doi:10.3390/nano11020420_

Round 1

Reviewer 1 Report

The manuscript entytled "3D Printing Fabrication of Personalized Nanomedicine" which is intended to be a review, has an interesting and actual interest in nanomedicine. Even so, the authors did not convince me that the present manuscript is worthy of being published in this form. I recommend the authors revise the manuscript and resubmit it. Maybe, before resubmitting the paper "Ten simple rules for writing a literature review" will be checked. Further some comments/recommendations are also listed:

  • The reference 104 do not report any information about perfusable hollow tubes obtained from cross-linked bioink. The authors should revise this.
  • The conclusions contain references ([71, 91, 92]), this being unusual for an article, even review. The conclusions should reveal the opinion on the reported study of the authors.
  • The English of the manuscript is hard to understand. The authors are required to revise it.
  • What is the reason of underlying two sentences at page 7?
  • Addressing to "Masks devised using 3D printing technology for protection from COVID-19" in this review is a little bit inappropriate. Even if it is still a desired subject to be addressed, I recommend to add more information on the type of materials used in the 3-D printing of masks against COVID-19. Anyway, there is only one reference in the whole manuscript that deals with, while for the other subjects as implants, wound healing, tissue engineering and regeneration and so on are based on much many studies.
  • There is only one reference of one author of the present review referred to in the manuscript. May the authors to add more own experience to the review?

Author Response

Comment# 1: The manuscript entitled "3D Printing Fabrication of Personalized Nanomedicine" which is intended to be a review, has an interesting and actual interest in nanomedicine. Even so, the authors did not convince me that the present manuscript is worthy of being published in this form. I recommend the authors revise the manuscript and resubmit it. Maybe, before resubmitting the paper "Ten simple rules for writing a literature review" will be checked. Further some comments/recommendations are also listed:

Reply:

We are thankful to reviewer for considering our manuscript interesting and correctly identifying the interest.

As per the suggestion of reviewer we have thoroughly revised the manuscript carefully considering the “ten simple rules for writing a literature”

Comment # 2: The reference 104 do not report any information about perfusable hollow tubes obtained from cross-linked bioink. The authors should revise this.

Reply: Complied.

Figure legend has been corrected in revised manuscript as Figure 1. 3D printing based designing of perfusable vascular constructs using cross-linked bioink ……………..”

Comment # 3 : The conclusions contain references ([71, 91, 92]), this being unusual for an article, even review. The conclusions should reveal the opinion on the reported study of the authors.

Reply: We agree with learned reviewer and hence we removed the references from conclusion section in revised manuscript; however we included these references to cite literature for different approaches and techniques of 3D printing which we haven’t reviewed in this article as sufficient literature is already available on this topic, in the favour of readers. 

Comment # 4 - The English of the manuscript is hard to understand. The authors are required to revise it.

Reply: We have revised entire manuscript carefully for English. 

Comment # 5 - What is the reason of underlying two sentences at page 7?

Reply - This was a formatting error which has now been corrected in revised manuscript. 

Comment # 6 - Addressing to "Masks devised using 3D printing technology for protection from COVID-19" in this review is a little bit inappropriate. Even if it is still a desired subject to be addressed, I recommend to add more information on the type of materials used in the 3- D printing of masks against COVID-19. Anyway, there is only one reference in the whole manuscript that deals with, while for the other subjects as implants, wound healing, tissue engineering and regeneration and so on are based on much many studies. 

Reply - Thank you for the valuable suggestion and accordingly we have added more information on 3D printing mask against COVID-19 in the revised manuscript and also included more relevant references.

Comment # 7 - There is only one reference of one author of the present review referred to in the manuscript. May the authors to add more own experience to the review?

Reply - Suggestion incorporated Required additions are done accordingly in revised manuscript. 

Reviewer 2 Report

Please kindly find below my comments regarding the review paper you sent me before:

The manuscript entitled “3D Printing Fabrication of Personalized Nanomedicine" was evaluated. The topic of the present review is novel and represents interesting data about progresses made in the field of personalized nanomedicine using 3D printing technology. I think it could be considered for publication in the “Nanomaterials", meanwhile some comments and major corrections are needed as below:

Comment 1. An extensive revision of the English language and spelling rules is recommended. The manuscript is full of grammar and spelling errors that some of them are highlighted in yellow.

Comment 2. The First sentence of Abstract defines Nanomedicine as “application of nanotechnology in designing of pharmaceutical drug products”. But based on the literature Nanomedicine is the medical application of nanotechnology and ranges from the medical applications of nanomaterials and biological devices, to nanoelectronics biosensors. As a matter of fact, nanomedicine is defined as the use of nanomaterials for diagnosis, monitoring, control, prevention, and treatment of diseases (Tinkle et al., 2014). On the other hand, authors indicate other diverse applications of 3D Printing in nanomedicine sections (wound healing, implants, tissue regeneration etc.). So, this section must be rewritten.

Comment 3. There are some publications related to the topic that can be cited, e.g. Jariwala, S.H., et al., 3D Printing of Personalized Artificial Bone Scaffolds. 3D Printing and Additive Manufacturing, 2015. 2(2): p. 56-64.

Comment 4. In Table 3, it is better to mention the name of pharmaceutical companies (in a separate column).

Author Response

Comment - The manuscript entitled “3D Printing Fabrication of Personalized Nanomedicine" was evaluated. The topic of the present review is novel and represents interesting data about progresses made in the field of personalized nanomedicine using 3D printing technology. I think it could be considered for publication in the “Nanomaterials", meanwhile some comments and major corrections are needed as below:

Reply - We are thankful to the reviewer for his positive remarks as well as comments which helped us in improving the quality of our manuscript. 

Comment 1. An extensive revision of the English language and spelling rules is recommended. The manuscript is full of grammar and spelling errors that some of them are highlighted in yellow.

Reply - Manuscript has been revised carefully for spelling and English usage. Since we didn’t upload the manuscript in the Nanomaterials MDPI journal format, and hence when it was converted to that format some of the errors may have appeared.  

Comment 2. The First sentence of Abstract defines Nanomedicine as “application of nanotechnology in designing of pharmaceutical drug products”. But based on the literature Nanomedicine is the medical application of nanotechnology and ranges from the medical applications of nanomaterials and biological devices, to nanoelectronics biosensors. As a matter of fact, nanomedicine is defined as the use of nanomaterials for diagnosis, monitoring, control, prevention, and treatment of diseases (Tinkle et al., 2014). On the other hand, authors indicate other diverse applications of 3D Printing in nanomedicine sections (wound healing, implants, tissue regeneration etc.). So, this section must be rewritten.

Reply - We are thankful to the learned reviewer and accordingly we have revised the abstract.  

The section “applications of 3D Printing in nanomedicine sections have been appropriately modified in revised manuscript. 

Comment 3. There are some publications related to the topic that can be cited, e.g. Jariwala, S.H., et al., 3D Printing of Personalized Artificial Bone Scaffolds. 3D Printing and Additive Manufacturing, 2015. 2(2): p. 56-64.

Reply - Suggestion incorporated

The suggested reference was directly related to the theme of this article and hence accordingly this and some other relevant references have been cited in revised manuscript.

Comment 4. In Table 3, it is better to mention the name of pharmaceutical companies (in a separate column).

Reply - Table 3 presents the summary of different 3D printing technologies being investigated for pharmaceutical formulation development. Various pharmaceutical companies like Aprecia Pharmaceuticals, GlaxoSmithKline etc are investigating 3D-printing technology to transform the pharmaceutical industry. Since in this table we also included the research work being carried out by scientists to develop 3D printed pharmaceutical formulation hence accordingly we have modified the table caption.

Reviewer 3 Report

Some sentences need to be rewritten for clarity, and typos need to be corrected, so the entire text needs to be carefully revised. Chapter 1.1 does not clarify the connection between 3d printing and nanomedicine. COVID-19 can be removed from the keywords because 3d printing has only an indirect connection. The title "3D Printing for Fabrication of Personalized Medicine and Nanomedicine" is better.

Author Response

Comment # 1: Some sentences need to be rewritten for clarity, and typos need to be corrected, so the entire text needs to be carefully revised. 

Reply - We have carefully revised the entire manuscript in this regard. 

Comment # 2 - Chapter 1.1 does not clarify the connection between 3d printing and nanomedicine

Reply - We have modified the section 1.1 to clarify the connection between 3D printing and Nanomedicine.

Comment # 3 - COVID-19 can be removed from the keywords because 3d printing has only an indirect connection.

Reply - Complied. Necessary change has been done in revised manuscript. 

Comment # 4  - The title "3D Printing for Fabrication of Personalized Medicine and Nanomedicine" is better.

Reply - After considering the suggestions of reviewer 3 and reviewer 4 we have modified the title of manuscript as “3D Printing in Development of Nanomedicines and Precision Medicines”. We used the term Precision medicine in place of older term “personalized medicine” as it could be misinterpreted to imply that treatments and preventions are being developed uniquely for each individual; whereas precision medicine focus on identifying which approaches will be effective for which patients based on genetic, environmental, and lifestyle factors (What is the difference between precision medicine and personalized medicine? What about pharmacogenomics? Accessed from: https://medlineplus.gov/). 

Reviewer 4 Report

The authors described a spectrum of biomedical applications using 3D printing technique, which is good and important for biomedical engineering. Nonetheless, the content of the present review is out of the title of 3D printing fabrication of personalized nanomedicine. I did not observe the authors were trying to summarize the progress of 3D printing in fabrication of biomedicine in nanoscale or even microscope, not to mention personalized nanomedicine. Hence, if the authors want to get this review published, I suggest to rephrase the title and reconstruct abstract.

Author Response

Comment - The authors described a spectrum of biomedical applications using 3D printing technique, which is good and important for biomedical engineering. Nonetheless, the content of the present review is out of the title of 3D printing fabrication of personalized nanomedicine. I did not observe the authors were trying to summarize the progress of 3D printing in fabrication of biomedicine in nanoscale or even microscope, not to mention personalized nanomedicine. Hence, if the authors want to get this review published, I suggest to rephrase the title and reconstruct abstract.

Reply - We greatly appreciate the valuable comments of the learned reviewer which have now been incorporated in the revised version of our manuscript. As suggested by the reviewer, the title and the abstract of article are rephrased suitably in the revised manuscript. 

Round 2

Reviewer 1 Report

Although the authors revised the manuscript in an attempt to improve it, the quality of the provided review is still of low quality, in comparison with other review reports in the field. Even with the modified title, the manuscript does not report a clear, synthetic line of the implication of 3D printing in nanomedicine and precision medicine.  Is to hard to follow how 3D printing helps in developing nanomedicine and precision medicine at the same time. I would suggest to focus on only one direction of two. As well, there is to hard to distinguish the 3D printing of nanomaterials and the general 3D printing used in development of nanomedicine and precision medicine. The manuscript can not be considered for publication in the present form. It should be improved.

Author Response

Reviewer 1

Comment

Although the authors revised the manuscript in an attempt to improve it, the quality of the provided review is still of low quality, in comparison with other review reports in the field. Even with the modified title, the manuscript does not report a clear, synthetic line of the implication of 3D printing in nanomedicine and precision medicine.  Is to hard to follow how 3D printing helps in developing nanomedicine and precision medicine at the same time. I would suggest to focus on only one direction of two. As well, there is to hard to distinguish the 3D printing of nanomaterials and the general 3D printing used in development of nanomedicine and precision medicine. The manuscript can not be considered for publication in the present form. It should be improved.. 

Reply

As per the suggestions of the learned reviewer we have revised the title of manuscript as “3D Printing in Development of Nanomedicines” to make it clear. Further we have revised the manuscript to make a clear, synthetic line of the implication of 3D Printing in nanomedicine and precision medicine. Agreeing to valuable suggestion of reviewer we have focused on only one direction in revised manuscript and in one section we have discussed the implications of 3D printed medicines in precision medicines.

Reviewer 2 Report

Despite the provided revision, In my opinion, the review is of poor quality and its distinction to many review papers that are already available is not clear. the authors need to make a clear line between 3D printing and 3D printing of nanomaterials. in my opinion the paper doesn't seem yet focus enough or different enough to be publishable. 

Author Response

Comment

Despite the provided revision, In my opinion, the review is of poor quality and its distinction to many review pape are already available is not clear. The author need to make it clear line between 3D printing and 3D printing of nanomaterials. In my opinion, the paper does not seem yet focus enough to be publishable. 

Reply

To answer the query of learned reviewer we would like to highlight the parts in our review article which makes it distinct from already available review papers including 3D printing in thermoresponsive hydrogels, tissue engineering and regeneration, precision medicines and regulatory perspectives related to 3D printed medicines. The revised manuscript is more focused on applications of 3D printed fabrication of nanomaterials for biomedical applications as nanomedicines. As per valuable suggestions of reviewer the thorough revision of manuscript is already done and revised manuscript is concentrated specifically to application of 3D printing in nanomaterials, we have highlighted in various places of revised manuscript in this regard. Authors highly thankful to reviewer for this comment to enhance the overall quality of this review article.

Reviewer 3 Report

The authors of the paper completed and corrected the manuscript in according to reviewers' observations.

Author Response

Reviewer 3

Comment

The authors of the paper completed and corrected the manuscript in according to reviewers' observations. 

Reply

We are thankful for positive comments of reviewer.  

Reviewer 4 Report

No further revision is needed

Author Response

Reviewer 4

Comment

No further revision is needed

Reply

Thank you 

Round 3

Reviewer 1 Report

The manuscript has been improved and now can be considered for publication in Journal Nanomaterials.

Reviewer 2 Report

The authors sufficiently replied to my comments.